# Survey of Serum Amyloid A and Bacterial and Viral Frequency Using qPCR Levels in Recently Captured Feral Donkeys from Death Valley National Park (California)

**DOI:** 10.3390/ani10061086

**Published:** 2020-06-23

**Authors:** Sara Jerele, Eric Davis, Samantha Mapes, Nicola Pusterla, Francisco Javier Navas González, Carlos Iglesias Pastrana, Essam Mahmoud Abdelfattah, Amy McLean

**Affiliations:** 1International Animal Welfare Training Institute, School of Veterinary Medicine, University of California Davis, Davis, CA 95617, USA; jerele.sara@gmail.com (S.J.); ewdavis@ucdavis.edu (E.D.); 2Equine Viral Disease Laboratory, Department of Pathology, Microbiology, and Immunology, School of Veterinary Medicine, University of California Davis, Davis, CA 95617, USA; smmapes@ucdavis.edu (S.M.); npusterla@ucdavis.edu (N.P.); 3Department of Genetics, Faculty of Veterinary Sciences, University of Córdoba, 14071 Córdoba, Spain; ciglesiaspastrana@gmail.com; 4The Worldwide Donkey Breeds Project, Faculty of Veterinary Sciences, University of Córdoba, 14071 Córdoba, Spain; 5Department of Animal Hygiene, and Veterinary Management, Faculty of Veterinary Medicine, Benha University, Qalubiya Governorate 13511, Egypt; essam.abdelfattah@fvtm.bu.edu.eg; 6Department of Animal Science, University of California Davis, Davis, CA 95616, USA

**Keywords:** donkeys, burro, serum amyloid A, *Streptococcus equi*, Asinine Herpesvirus, PCR

## Abstract

**Simple Summary:**

This study aimed to measure the inflammatory marker (Serum Amyloid A; SAA) derived from such pathogenic processes specifically bacterial and virus found in recently captured feral donkeys. Quantitative Real-Time PCR (qPCR) was performed to detect and quantify RNA and DNA viruses and bacterial DNA. Behavior, body condition score, nasal discharge, and coughing were found to be related in cases of Asinine Herpesvirus 2 (AHV-2) and *Streptococcus equi* subspecies *zooepidemicus* DNA. SAA concentrations descended with age, with foals presenting higher concentrations. Positive tests differed for AHV-2 and *Streptococcus equi* spp. *zooepidemicus* between sampling moments. In conclusion, donkeys caught in the wild may not be a source of disease for domestic horses. However, the transmission of some pathogens, such as *Streptococcus equi* subspecies *zooepidemicus,* and/or AHV-2, may occur.

**Abstract:**

Feral donkey removal from state land has raised concerns in terms of disease transmission between equine species. Disease outbreaks may occur as a result of the relocation of animals to new environments. Virus and bacteria DNA load and serum amyloid A derived from the pathogenic processes that they involve were measured in recently captured donkeys. Blood and nasal swabs were collected from 85 donkeys (Death Valley National Park, Shoshone, California); 24 were retested after 30/60 days in the Scenic (Arizona) long-term holding facility co-mingled with feral donkeys from Arizona and Utah. Quantitative Real-Time PCR (qPCR) was performed to detect viral and bacterial genomic material (equine influenza A [EIV], equine rhinitis A and B viruses, AHV-2, AHV-3, AHV-5 and EHV-1, EHV-4, *Streptococcus equi* subspecies *equi* and *zooepidemicus*,). Significant relations between behavior, body condition score, nasal discharge, and coughing were found in donkeys for which AHV-2 and *Streptococcus zooepidemicus* DNA was detected. Higher SAA concentrations were found in foals. AHV-2 and *Streptococcus zooepidemicus* DNA concentrations significantly differed between sampling moments (*p* < 0.05). In conclusion, donkeys do not appear to be a substantial risk for disease transmission to horses but could be if they carried strangles or other processes in which AHV-2 and *Streptococcus zooepidemicus* were involved.

## 1. Introduction

The feral burro or donkey population of Death Valley presumably originates from the descendants of domesticated donkeys that were brought to the southwestern deserts by miners in the USA over 100 years ago [1]. These hardy animals found an ideal environment in Death Valley, with enough feed and water and no natural predators. In the past century, the burro has become part of the megafauna of this inhospitable area, though some believe their effect on native flora and biota is deleterious [2]. Like the Bureau of Land Management (BLM) on other state lands, the National Parks Service (NPS) has been tasked with the management of all wild animals living in the Death Valley National Park. As part of their mission, the NPS assesses burro (donkey) populations and has made the managerial decision to reduce herd sizes with a nonprofit capture and relocate the burros [3]. 

As little is known about the overall health of feral donkeys and the prevalence of pathogens, specifically in the category of infectious equine viruses and bacteria in their populations [4]; it is critical to conduct field health assessments of donkeys to predict early potential diseases before relocation. In this regard, Serum Amyloid A (SAA), is an acute-phase apolipoprotein, synthesized by the liver as a response to inflammation derived from infection or injury [5,6,7]. A new SAA test kit provides rapid on-site SAA blood concentrations (StableLab, Sligo, Ireland), which may be of help when assessing the health condition of feral donkeys or in field conditions. Normal concentrations of plasma SAA are virtually undetectable [7]; rapid increasing concentrations of SAA, from ten to 1000 times that of normal levels, indicate some form of inflammation or infection [8]. Countries such as the USA have run surveillance programs for equine infectious respiratory pathogens [9]. These programs focus on the prevalence and epidemiology of important viral and bacterial agents, such as equine influenza virus (EIV), equine herpesvirus-1 and -4 (EHV-1 and EHV-4), and *Streptococcus equi* subspecies *equi* in horses (*Equus caballus*). However, such studies have not been performed on donkeys. 

Contextually, *Streptococcus equi* subspecies *zooepidemicus* (*S. zooepidemicus*) and subspecies *equi* (*S. equi*) infect equids worldwide but are especially prevalent in donkeys [10]. *S. zooepidemicus*, a commensal organism in the equine nasopharynx, can occasionally invade the respiratory mucosa, resulting in purulent rhinitis, bronchitis, lymphadenopathy, and pneumonia in horses and donkeys of all ages [10,11]. Furthermore, single strains with *S. zooepidemicus* phenotypes similar to those found in the nasopharynx of healthy horses have been reported to cause pneumonia in donkeys [12]. Donkeys have been reported to act as asymptomatic chronic carriers of *S. zooepidemicus*, with conditions being characterized by guttural pouch infection or intermittent nasal discharge. Indeed, clinical observations [13] have suggested an increased likelihood of guttural pouch disease in asses. 

The *herpesviridae* family plays an important role in the prevalence and epidemiology of equid respiratory disease. This viral family sub-divides into the *alpha-* (EHV-1, -3, and -4) [14], *beta-,* and *gamma-* (EHV-2 and -5) *herpesviridae* which can infect all equids [15]. Donkeys and hybrids (mules and hinnies) may not be a source of herpesviruses transmission to other equids [16], but host a range of specific Asinine Herpesviruses [17]. Asses are susceptible to EHV-1 and -4, with respiratory and abortion being described in the literature. Three of the herpesviruses found in donkeys have been isolated and identified as similar to EHV-1, -7, and -8 [11,12,13,14,15,16,18] and have been described and designated as both equid (EHV) and Asinine Herpesviruses (AHV) [18]. Asinine Herpesvirus 2 (AHV-2), also denoted as EHV-7 [18], closely resembles gamma-herpesviruses EHV-2 and -5 [16], can cause clinical signs of respiratory infection, and can also be isolated from clinically healthy donkeys [18]. 

Limited information is available regarding AHV-3 (designated as EHV-8) [17,18] AHV-4, AHV-5 [19], and AHV-6 [20]. The GG nucleotide sequences-assisted phylogenetic assessment addressed a shorter evolutionary distance between AHV-3 and EHV-1 than between either virus or EHV-4. These outcomes contribute to the hypothesis that AHV-3, or other affine or related viruses, are the precursors of EHV-1 and only adapted to horses relatively recently [17]. 

Thiemann [21] described AHV-5 isolates from lungs displaying interstitial pneumonia and remarkable syncytial cell formation which did not affect lung tissue in donkeys, which presented evidence of pneumonia with a bacterial or verminous underlying etiology [19]. Furthermore, this gamma herpesvirus was similar to that of a donkey identified with neurological disease [22]. AHV-4 and -5 isolates have been obtained from lungs belonging to donkeys coursing pulmonary fibrosis. Donkey’s pulmonary fibrosis is similar to horses’ multinodular pulmonary fibrosis, which has been ascribed to the action of EHV-5. Positive cases of alpha herpesviruses EHV-1 and -4 from animals presenting respiratory and neurological conditions have been clinically described as well [21]. 

Stress from relocation and social comingling present an increased likelihood for outbreaks of latent infections [23]. Typically described clinical signs, such as a harsh cough, pyrexia, serous nasal secretion, and lethargy have been reported, as also described for horses. Donkey herpesviruses may differ from that of horses, although the guidelines for diagnosis, treatment, and control remain the same [21]. However, horse EHV-1 and EHV-4 vaccination viability have not been reported in donkeys. Coinfection of some of *S. zooepidemicus* group C and Asinine Herpesvirus 5 (AHV-5) with *Dictyocaulus arnfieldi* has been hypothesized to contribute to the clinical disease, but without empirical or experimental evidence [24]. 

This study aimed to measure the frequency of potential pathogens in recently captured feral donkeys from Death Valley National Park, testing for the potentially existing correlations of pathogen loads with respiratory, ocular, behavioral, ambulatory clinical signs and inflammatory markers (serum amyloid A) to assess whether changes in pathogen presence after donkeys are co-mingled with domestic equids is warranted.

## 2. Materials and Methods 

### 2.1. Animal Sample and Sampling

Whole blood and nasal discharge were sampled from 85 feral donkeys (37 jennies and 48 jacks) removed from Death Valley National Park range in Shoshone, California (GW4C+56 Skidoo, California, United States/ 36°29′58.7″ N 117°03′07.7″ W). A 12 mL blood sample was collected by venipuncture of the jugular vein by syringe for further analysis. Nasal secretions were collected on a synthetic swab with a plastic applicator and stored in a small zip-lock plastic bag at −20 °C until extraction. All jennies and foals were moved to a long-term holding facility in Scenic, Arizona (176 miles away) and co-mingled with other wild donkey populations from Arizona and Utah (sampling moment one, 4–5 November 2018 (group one) and 4–5 December 2018 (group two)). Twenty-four donkeys were retested for SAA and nasal secretions (NS) (sampling moment two, around 30/60 days later, 7 January 2019). The second sampling (moment two) was set at 30/60 days to ensure a minimum period of 21 days had been left as a way to cover the incubation period of all potential pathogens present. Clinical examination was performed on all animals before sampling by two operators. Invasive procedures were limited to blood sampling and nasal swabs. Temperature Pulse Respiration (TPR) was not performed as abnormally elevated values could be expected. In this context, the correct assessment of health status could have been distorted, as a result of the lack of handling and human interaction of the feral donkeys participating in the study until the moment of the sampling. Visual examination comprised records on body condition score (BCS), gender (37 jennies and 48 jacks), age (81 adults of ≥2 years old, and 4 foals, 1 jenny foal and 3 male foals, which were still nursing), the presence of skin/hair conditions (absent, alopecia, photosensitivity/photodermatitis, minor wound, deep wound), the presence of lameness (1 lame, 2 not lame when visually observed walking from holding pen to chute), the presence of either ocular or nasal discharge (1 present and 2 not present), the presence of coughing (1 present if animal was coughing and 2 not present if animal was not coughing), the presence of abnormal breathing (1 if abnormal breathing was observed during sample collection and 2 if it was not present), and behavior signs (alert, lethargic, aggressive, fearful). BCS was determined following a 1 to 5 scoring system, 1 poor to 5 obese, described in Valle et al. [25] and Polidori and Vincenzetti [26] and represented in Figure 1. The presence of lameness was assessed through visual examination of the holding pen where they were kept to the area were these were sampled in the chutes. Since the donkeys were feral/wild animals, lameness was evaluated by only visual observation: 1 if donkey was lame and displayed signs of lameness like bobbing head, irregular gait, non-weight bearing limb were visually observed; or 2 if no irregular gait or obvious visual signs were observed by the clinicians. Behavior signs were evaluated following the premises in Navas González et al. [27] by visually observing the face (eye, nostril and ear position), head, neck and body posture and language. Cronbach’s alpha parameter to test for interobserver reliability reported agreement based on internal consistency was over 0.8 for the observations of the two operators performing a clinical examination, hence their information could be considered objective enough to be included in the study.

### 2.2. SAA Levels 

Whole blood was analyzed for serum amyloid A (SAA) by using a commercial stable-side kit (StableLab, Sligo, Ireland). Whole blood was taken directly from the syringe and no additive was added. SAA concentrations were obtained from the kit after 10 min post-sampling on-field, and the values for SAA concentrations expressed in mg/L were recorded. Further information on the procedures used to test for SAA can be found in Kay et al. [7].

### 2.3. Load Quantification for Viruses and Bacteria

Nasal swabs were collected from the left nostril as the position in which the animal was restrained made it more accessible. Immediately after collection, blood samples were collected and transferred to the lab in portable refrigerators. Nasal swabs were collected in virus transport media (VTM) and stored at −80 °C until testing. The median time-lapse from collection to processing was 2 days (Interquartile range 2–3 days). Additional information about sample preparation, submission, and process can be found in the q-PCR diagnostic submission packet (University of California Davis’ Veterinary Medicine PCR Laboratory [28]).

Once at the lab, swabs were placed in a 15 mL conical with 500 μL phosphate-buffered saline, vortexed and spun at 4000 rpm to obtain a cell pellet. Total nucleic acid was extracted from the swabs using a QIAcube HT automated nucleic acid (ANA) workstation (Qiagen, Valencia, CA, USA) according to the manufacturer’s instructions for the QIAamp 96 DNA QIAcube HT Kit (Qiagen). 

Real-time PCR (quantitative PCR, qPCR) was performed using nasal secretions for the detection and quantification of different microbial agents: *Streptococcus equi* subspecies *equi, Streptococcus equi* subspecies *zooepidemicus,* equine influenza type A (H3N8), equine Rhinitis A, equine Rhinitis B, AHV-2, AHV-3, AHV-5, Equine Herpesvirus 1, Equine Herpesvirus 1 (EHV-1, neuropathogenic), Equine Herpesvirus- 1 EHV-1(non-neuropathogenic), and Equine Herpesvirus-4 (EHV-4).

### 2.4. Quantitative PCR Systems

Primer Express software (Thermo Fisher Scientific, Carlsbad, CA, USA) was used to design two primers and an internal hydrolysis fluorescent-labeled probe (5′ end, reporter dye fluorescein amidite (FAM) (6-carboxyfluorescein), 3′ end, quencher NFQMGB (Non-Fluorescent Quencher Minor Grove Binding) for each target gene (Table 1). Unique species detection was confirmed by a Basic Local Alignment Search Tool (BLAST) of each amplicon. Ten-fold dilutions of DNA testing positive for the target genes were used to validate qPCR (real-time or quantitative polymerase chain reaction) systems. The dilutions were analyzed in triplicate and a standard curve plotted against the dilutions. Amplification efficiencies were computed through the formula E = 10 ^1/-s^-1 (slope of the standard curve). A minimum of 90% efficiencies was required to pass validation. 

### 2.5. RT-Reaction and Quantitative PCR

The cDNA synthesis was performed following the modified protocol of manufacture prescriptions of the Quantitect Reverse transcription kit (Qiagen). Ten microliters of RNA were digested with 1 μL of gDNA wipeout buffer by incubation at 42 °C for 5 min and then briefly centrifuged. A total of 1 μL of digested RNA and the real-time PCR housekeeping gene was used to test for genomic DNA contamination. Then, 0.5 μL of Quantitect Reverse Transcriptase, 2 μL Quantitect RT (Real-Time) buffer, 0.5 μL RT primer mix, 0.5 μL 20 pmol random primers (Invitrogen) were added and brought up to a final volume of 20 μL and incubated at 42 °C for 40 min. The samples were inactivated at 95 °C for 3 min, chilled, and 80 μL of water was added. 

Each PCR reaction contained 20× primer and probes for the respective qPCR system, with a final concentration of 400 nM for each primer and 80 nM for the probe and commercially available PCR master mix (TaqMan Universal PCR Mastermix, Applied Biosystems) containing 10 mM Tris-HCl (pH 8.3), 50 mM KCl, 5 mM MgCl_2_, 2.5 mM deoxynucleotide triphosphates, 0.625 U AmpliTaq Gold DNA polymerase per reaction, 0.25 U AmpErase UNG per reaction and 5 μl of the diluted gDNA and cDNA sample in a final volume of 12 μL. The samples were placed in 384 well plates and amplified in an automated fluorometer (ABI PRISM 7900 HTA FAST, ABI). ABI’s standard amplification conditions were used: 2 min at 50 °C, 10 min at 95 °C, 40 cycles of 15 s at 95 °C and 60 s at 60 °C. Fluorescent signals were collected during the annealing temperature and CT (cycle threshold) values extracted with a threshold of 0.1 and baseline values of 3–12. 

### 2.6. Statistical Analyses

Field observations were collected at two moments (sampling moments one and two). The population membership changed over time from sampling moment one to sampling moment two but retained certain common participants, which are known as partially overlapping samples. As suggested by Derrick et al. [29], the assumptions of partially overlapping samples t-test with Welch’s degrees of freedom are the same as those for Welch’s test (normality and heteroscedasticity [30]). When a non-gross violation of the normality assumption has occurred, Welch’s test power could be comparable to t-test power if homoscedasticity is still met, as suggested Rasch et al. [31]. Still, in cases of unequal variances and skewness values < 3, Welch’s test has been reported to keep 20% robustness. Parametric assumptions were tested to determine the best set of tests to statistically process data. All parametric assumption testing tests, except for the normality tests, were performed using SPSS Statistics for Windows, Version 25.0, IBM Corp. [32]. Normality assumptions were tested using StataCorp Stata version 14.2 one-way ANOVA routine of the compare samples task of SPSS Statistics for Windows, Version 25.0, IBM Corp. [32], and partially overlapping samples t-test with Welch’s degrees of freedom using the “Partially overlapping” package by Derrick [33] of RStudio 1.1.463 by the R Studio Team [34] were used to detect differences in the mean of viral and bacterial DNA quantities. 

Pearson correlations were used to determine the relationship between SAA maximal concentration expressed in mg/L and microbial DNA quantity using the Bivariate procedure from the correlate task from SPSS Statistics for Windows, Version 25.0, IBM Corp. [32]. Afterward, categorial regression (CATREG) was performed to determine best-fitting linear predictive models for SAA maximal concentrations and AHV-2 and *S. zooepidemicus* DNA quantities using combinations of the five significant independent factors (sampling moment, BCS, behavior, nasal discharge presence, and coughing). Categorical regression was performed using the Optimal Scaling procedure from the Regression task from SPSS Statistics for Windows, Version 25.0, IBM Corp. [32].

### 2.7. Ethical Approval

This study was approved by the University of California Davis’ Animal Care and Use Committee IACUC #20611.

## 3. Results

Parametric assumptions were tested to decide whether a parametric or nonparametric approach should be implemented. Homoscedasticity (Levene’s test for equality of error variance, *p* < 0.05) assumption for DNA quantity across the different categories for all the factors studied (respiratory, ocular, behavioral, ambulatory clinical signs, and SAA maximal approximate concentration) was violated. Normality was assumed (Shapiro–Wilk’s Francia W, *p* > 0.001) as our sample was obtained out of a presumably normal population, hence a partially overlapping samples t-test with Welch’s degrees of freedom was carried out to determine the differences in the means across factor levels for each of the pathogens. Summaries of the descriptive statistics for both sampling moments one and two are reported in Table 2 and Table 3, respectively.

The summary of the results for the partially overlapping samples t-test to determine the differences in the means in levels for clinical signs, the SAA maximal concentration expressed in mg/L, qPCR microbial load detection, and controls between sampling moment one and two are shown in Appendix A. Significant differences across the levels for BCS, behavior signs, nasal discharge, and coughing presence were found (*p* < 0.05) between sampling moments one and two (Appendix A). In these regards, for pathogens quantification and controls between sampling moments one and two, significant differences were found for AHV-2 aka EHV7, and *Streptococcus equi* subspecies *zooepidemicus.*

The BCS reported a 0.69 higher score in sampling moment one compared to sampling moment two. Behavior signs reported a 0.15 higher score, which translated into a higher trend to display alert or fearful responses; nasal discharge reported a significant increased higher frequency of 0.24 in sampling moment one, while this increased higher trend was reported for coughing, which was more frequent in sampling moment one than two (0.80 value for estimate difference in the means, as shown in Appendix A). When the qPCR results were compared from the initial sampling to the final sampling, there was a significant increase in the mean of 1.66 and 2.00, for AHV-2 aka EHV7 and Streptococcus *equi* subspecies *zooepidemicus*, respectively.

Afterward, a CATREG (Categorical Regression)was performed using five independent significant factors in Appendix A (sampling moment, body condition score, behavior, nasal discharge presence, and coughing) as predictors and AHV-2 and *Streptococcus zooepidemicus* qPCR findings as the dependent variables. Appendix A reports a model summary of stepwise linear regression for AHV-2 and *Streptococcus equi* subspecies *zooepidemicus* qPCR-assay. The standardized coefficients (β) for predictive factors are listed in Appendix A and were used to build predictive equations. 

The CATREG reported that none of the factors was linearly related with AHV-2 load. However, *Streptococcus equi* subspecies *zooepidemicus* significantly depended on the independent variables of behavior, coughing presence, and sampling moment. Concretely, the CATREG reported the model predicting for AHV-2, explained 20.5% of the variability in qPCR quantifications for such pathogen (Appendix A). Considering the value of the bootstrap (1000) estimate of standard error, we could identify a nonlinear monotonic relationship between AHV-2 qPCR pathogen load and nasal discharge presence. 

The model predicting for *Streptococcus equi* subspecies *zooepidemicus* was able to capture 15.2% of the variability in qPCR results (Appendix A). In this case, and considering the value of the bootstrap (1000) estimate of standard error, we could identify a significant linear monotonic relationship between *Streptococcus equi* subspecies *zooepidemicus* PCR quantities and the behavior, coughing and sampling moment. The resulting predictive regression equation for *Streptococcus equi* subspecies *zooepidemicus* qPCR detection is as follows: Zy’_Streptococcus equi subspecies zooepidemicus_ = β_behavior_ * Z_behavior_ + β_coughing_ * Z_coughing_ + β_samplingmoment_ * Z_samplingmoment_(1)
where Zy’_*Streptococcus equi* subspecies *zooepidemicus*_ is the phenotypic record of *Streptococcus equi* subspecies *zooepidemicus* PCR quantities, β is the standardized coefficient for each independent factor as marked by a sub-index for the whole population, and Z is the specific value for that same factor for each individual. β is the standardized coefficients and can be found in Appendix A. Only significant coefficients were considered for the regression equation as they identified the variable which holds a linear relationship with the amount of DNA tested by qPCR. The analysis of Pearson correlations reported a significant linear correlation between age and the SAA maximal approximate concentration of 0.561 (*p* < 0.001), while the rest of the factors seemed not to be correlated to SAA maximal approximate concentration expressed in mg/L.

## 4. Discussion

Donkeys in general are stoic animals that often do not respond in the same manner to disease as horses. When working with feral donkey and attempting to diagnose the health status without diagnostic tools or the ability to fully perform clinical exams, the methodology used in this study may provide an adequate approach to identifying healthy and/or sick animals. Donkeys gathered and relocated to short-term and or long-term facilities will likely go through increased stress. During the relocation process to new environments, incoming donkeys co-mingling with many animals for multiple locations may further expose donkeys to pathogens. This exposure in a stressed state may cause disease outbreaks [35]. Jennies relocated to a long-term holding facility and co-mingled with donkeys from multiple locations showed an increase in AHV-2 and *Streptococcus zooepidemicus*. Equine herpesviruses (EHVs) can affect all members from the *Equidae* family by establishing acute [36] and latent infections [37]; hence, transmission does not face any totally effective preventive interspecific barrier. Over an animal’s lifetime, latent herpes infections can reoccur or new acute infections can occur [36].

Generally, few clinical signs of infection or suggested guidelines for prevention and control in donkeys have been made [36] for Asinine Herpesviruses. However, one study suggested that donkeys subjected to stress (e.g., relocation or transport) are at higher risk of the recrudescence of latent infections [38]. The use of such stable side tests like SAA may assist in detecting such disease at an early stage. In this study, we found an increase in SAA in young donkeys less than a year of age along with an increase in *Streptococcus zooepidemicus* and AHV-2 after being moved to long-term holding.

Our findings also suggested that nasal discharge presence was the most relevant sign to be considered given its direct relationship with an increased AHV-2 pathogen DNA quantity. The present information provides further insight into the recognition of AHV-2 in previously undiagnosed donkeys. AHV-2 (EHV-7) and other barely characterized gamma-herpesviruses have been isolated from asymptomatic healthy equids (horses, donkeys and mules), and from donkeys presenting encephalitis or severe interstitial pneumonia conditions [39].

The detection of AHV-5 from respiratory fluids from nasal swabs of horses that did not present any clinical sign of respiratory disease has been described [40,41]. Similarly, the asymptomatic donkeys in this study could test positive for AHV-5. Presumably, they have similar infection and latency patterns as those reported for EHV-2 and EHV-5 [41]. Goehring [42] reported EHV-2 to be present in the respiratory fluids and PBMCs (peripheral blood mononuclear cells) of the majority of foals from birth to weaning. These findings were in conjunction with moderate clinical disease symptoms such as fever, nasal discharge, pharyngeal follicular hyperplasia, and mandibular lymphadenopathy. These symptoms may agree with our findings, which address nasal discharge as the most relevant sign to be found in cases of EHV-2 infection. Similarly, when both EHV-2 and EHV-5 were found, the disease presented a rather severe course, as also suggested by our positive results.

In this group of animals, *Streptococcus equi* ssp. *zooepidemicus* genome quantities were latent for a longer time, potentially making this commensal bacterium more prone to the active transmission when feral animals are introduced with other equids. Coughing has been reported to be the most relevant sign and to be linearly correlated to *Streptococcus equi* ssp. *zooepidemicus* DNA quantity. The increased levels of coughing as bacterial loads increase may be also the basis for greater discomfort, which translates into the animal adopting a more alert or fearful status as reported by our study, while appetite appeared unaffected. *Streptococcus equi* ssp. *zooepidemicus* has been considered an opportunistic commensal in horses, though it has also been identified in the guttural pouches of systemically ill donkeys with empyema and chondroids [43]. This may suggest the role of *Streptococcus zooepidemicus* as a primary pathogen in donkeys. Contextually, our results suggest that this bacterium may be more important when recently feral donkeys are introduced to domestic equines.

Some authors [44] have even provided evidence supporting the fact that some highly frequent beta-hemolytic streptococci other than *S. equi,* such as *S. equisimilis* and *S. zooepidemicus,* could be the causative agent of strangle-like disease also associating its presence with other epidemic upper respiratory diseases. This contradicts Timoney [10], who reported that the related species *S. equi* is more virulent and much more likely to cause lymphadenopathy and abscessation (“strangles”) [10].

Donkeys and mules may be more susceptible to strangles with *S. zooepidemicus* acting as a primary pathogen, with a variable clinical presentation ranging from fever, purulent nasal discharge, and the abscessation of mandibular and retropharyngeal lymph nodes to mild persistent nasal discharge, as supported by our results. In this context, the first description of a strangles outbreak in donkey intensive farming systems was described in China in 2018 [45].

## 5. Conclusions

In conclusion, captured feral donkeys harbor some asinine viruses and *Streptococcus zooepidemicus* but not pathogens that might potentially infect other equine species. The comingling of wild-caught feral donkeys with domestic animals in a holding facility increases the levels of detectible *Streptococcus zooepidemicus* and AHV- 2 in the population, suggesting that captured wild donkeys may be at risk in holding facilities and during transport. Nasal discharge and coughing may be symptoms of the presence of potentially pathogenic organisms such as *Streptococcus zooepidemicus* and AHV- 2. The use of diagnostic testing tools such as stable side SAA kits and testing for pathogens through PCR nasal swab samples along with basic clinical exams may aide in the early detection of the disease and overall improve the health of donkeys. This study has contributed to our knowledge about SAA levels in healthy donkeys but additional research measuring SAA in clinically ill animals would be beneficial.

## Figures and Tables

**Figure 1 animals-10-01086-f001:**
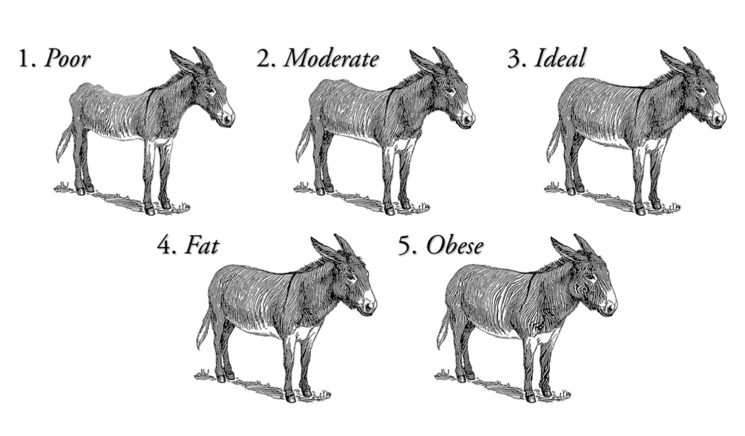
Body condition score in donkeys as described in Valle et al. [25] and Polidori and Vincenzetti [26].

**Table 1 animals-10-01086-t001:** Target genes used for qPCR-assay pathogen loads testing.

Assay Name	Gene, NCBI ^a^ Accession #	Assay Location (bp)	Amplicon Length (bp)
Asinine Herpesvirus 2	Polymerase, EU165547	100	81
Asinine Herpesvirus 3	Glycoprotein B, U24184	740	145
Asinine Herpesvirus 5	Polymerase, AY054993	600	64
Equine Herpesvirus 1	Glycoprotein B, NC_001491	400	89
Equine Herpesvirus 1, neuropathogenic	ORF 30, KF644574	200	92
Equine Herpesvirus 1, non-neuropathogenic	ORF 30, KX101095	200	92
Equine Herpesvirus 4	Glycoprotein B, AF030027	440	77
*Streptococcus equi* subspecies *equi*	M Protein, AF012927	150	185
*Streptococcus equi* subspecies *zooepidemicus*	ITS, EU860336	80	88
Influenza A (H3N8)	Hemagglutinin Precursor, EF541443	350	200
Equine rhinitis A virus	RNA polymerase, X96870	150	111
Equine rhinitis B virus	RNA polymerase, X96871	350	87
Glyceraldehyde-3-phosphate dehydrogenase	GAPDH, AF097179	60	105

^a^ NCBI: National Center for Biotechnology Information.

**Table 2 animals-10-01086-t002:** Summary of descriptive statistics for clinical examination signs, Serum Amyloid A (SAA) maximal concentration expressed in mg/L, qPCR-assay microbe loads, and controls at sampling moment one.

	Parameters	Mean	SEM	SD	Skewness	Kurtosis
Clinical Examination Signs	Body condition score (BCS)	3.12	0.09	0.83	−0.15	−0.96
	Behavior signs	2.11	0.04	0.38	1.14	3.13
	Skin/hair condition	1.29	0.10	0.94	3.21	9.14
	Lameness presence	1.96	0.02	0.19	−5.13	24.88
	Ocular discharge presence	1.82	0.04	0.38	−1.73	1.01
	Nasal discharge presence	1.66	0.05	0.48	−0.68	−1.57
	Abnormal breathing presence	1.07	0.03	0.26	3.41	9.88
	Coughing presence	1.93	0.03	0.26	−3.41	9.88
SAA	SAA maximal concentration (mg/L)	10.80	3.72	34.27	5.42	33.75
Pathogen Load qPCR-Assay	Asinine Herpesvirus 2 (AHV-2) aka EHV7	39.69	0.14	1.30	−4.33	18.30
	Asinine Herpesvirus 3 (AHV-3) aka EHV8	39.74	0.14	1.30	−5.92	36.83
	Asinine Herpesvirus 5 (AHV-5)	34.39	0.50	4.59	−0.51	−0.38
	Equine Herpesvirus 1 (EHV-1)	40.00	0.00	0.00	N/A	N/A
	Equine Herpesvirus 1 (EHV-1) neuropathogenic	40.00	0.00	0.00	N/A	N/A
	Equine Herpesvirus 1 (EHV-1) non-neuropathogenic	40.00	0.00	0.00	N/A	N/A
	Equine herpesvirus 4 (EHV-4)	40.00	0.00	0.00	N/A	N/A
	*Streptococcus equi* subspecies equi	39.94	0.04	0.39	−6.77	46.45
	*Streptococcus equi* subspecies *zooepidemicus*	35.04	0.44	4.08	-0.75	0.07
	Influenza AH3N8	40.00	0.00	0.00	N/A	N/A
	Equine rhinitis A virus	40.00	0.00	0.00	N/A	N/A
	Equine rhinitis B virus	40.00	0.00	0.00	N/A	N/A
Controls	Glyceraldehyde 3 phosphate dehydrogenase (First control)	29.24	0.38	3.49	0.83	1.91
	Glyceraldehyde 3 phosphate dehydrogenase (Second control)	32.13	0.41	3.78	0.27	−0.87
	Fibrinogen	0.44	0.09	0.50	4.56	23.39

SEM: Standard error of the mean; N/A: Not applicable as all animals reported the same values. Skewness Standard Error was 0.26 and Kurtosis Standard Error was 0.52 for all variables except for Fibrinogen, whose values were 0.41 and 0.81, respectively. BCS (1–5), Behavior signs (1 alert, 2 apathetic, 3 lethargic, 4 aggressive, 5 fearful, 6 withdrawn), Lesions (1 present, 2 not present), Lameness (1 present, 2 not present), Ocular discharge (1 present, 2 not present), Nasal discharge (1 present, 2 not present), Breathing (1 normal, 2 abnormal), coughing (1 present, 2 not present), SAA maximal concentration expressed in mg/L, qPCR-assay pathogen loads (40 = zero viral or bacterial agents present; lower numbers indicate higher number of DNA found), and controls at sampling moment one in 53 donkeys including jacks, jennies and foals.

**Table 3 animals-10-01086-t003:** Summary of descriptive statistics for clinical examination signs, SAA maximal concentration expressed in mg/L, qPCR-assay for organisms and controls at sampling moment two.

	Parameters	Mean	SEM	SD	Skewness	Kurtosis
Clinical Examination Signs	Body condition score	2.43	0.08	0.37	0.97	1.81
	Behavior signs	1.96	0.07	0.36	−0.65	6.34
	Skin/Hair condition	1.00	0.00	0.00	N/A	N/A
	Lameness presence	2.00	0.00	0.00	N/A	N/A
	Ocular discharge presence	1.87	0.07	0.34	−2.42	4.21
	Nasal discharge presence	1.42	0.10	0.50	0.36	−2.05
	Abnormal breathing presence	1.13	0.07	0.34	2.42	4.21
	Coughing presence	1.13	0.07	0.34	2.42	4.21
SAA	SAA maximal concentration (mg/L)	17.42	10.73	52.58	4.43	20.53
Pathogen Load qPCR-Assay	Asinine Herpesvirus 2 (AHV-2) aka EHV7	38.03	0.90	4.40	−2.57	6.18
	Asinine Herpesvirus 3 (AHV-3) aka EHV8	39.94	0.06	0.31	−4.90	24.00
	Asinine Herpesvirus 5 (AHV-5)	32.65	0.89	4.36	−1.37	2.84
	Equine Herpesvirus 1 (EHV-1)	40.00	0.00	0.00	N/A	N/A
	Equine Herpesvirus 1 (EHV-1) neuropathogenic	40.00	0.00	0.00	N/A	N/A
	Equine Herpesvirus 1 (EHV-1) non-neuropathogenic	40.00	0.00	0.00	N/A	N/A
	Equine Herpesvirus 4 (EHV-4)	40.00	0.00	0.00	N/A	N/A
	*Streptococcus equi* subspecies equi	40.00	0.00	0.00	N/A	N/A
	*Streptococcus equi* subspecies *zooepidemicus*	33.04	0.69	3.38	-0.65	0.77
	Influenza AH3N8	40.00	0.00	0.00	N/A	N/A
	Equine rhinitis A virus	40.00	0.00	0.00	N/A	N/A
	Equine rhinitis B virus	40.00	0.00	0.00	N/A	N/A
Controls	Glyceraldehyde 3 phosphate dehydrogenase (First control)	23.61	0.97	4.76	2.04	5.02
	Glyceraldehyde 3 phosphate dehydrogenase (Second control)	29.37	0.85	4.17	1.45	1.75
	Fibrinogen	0.63	0.54	2.67	4.73	22.74

N/A: Not applicable as all animals reported the same values. Skewness Standard Error was 0.47 and Kurtosis Standard Error was 0.92 for all variables. BCS (1-5), Behavior signs Behavior signs (1 alert, 2 apathetic, 3 lethargic, 4 aggressive, 5 fearful, 6 withdrawn), Lesions (1 present, 2 not present), Lameness (1 present, 2 not present), Ocular discharge (1 present, 2 not present), Nasal discharge (1 present, 2 not present), Breathing (1 normal, 2 abnormal), coughing (1 present, 2 not present), SAA maximal concentration expressed in mg/L, qPCR-assay microbe loads (40 = zero viral or bacterial present; lower numbers indicate higher number of DNA found), and controls at sampling moment two in 30 donkeys (28 jennies and 4 foals) in long-term holding facilities.

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
