# Peer review of "Survey of Serum Amyloid A and Bacterial and Viral Frequency Using qPCR Levels in Recently Captured Feral Donkeys from Death Valley National Park (California)"

_animals, 2020, doi:10.3390/ani10061086_

Round 1
Reviewer 1 Report
The authors presented a paper on serum amyloid A and pathogens qPCR levels in feral donkeys in California.
In my opinion the research is adequately conducted and the paper is well written.
I have only some minor comments to suggest before the acceptance for the publication.
General comment: in my opinion the introduction section is too long and strong to read. The authors furnished a lot of unnecessary information about viruses and bacteria. Please try to summarize this section.
More in details:
-if possible try to better differentiate the first lines in simple summary and abstract: lines 23-24-25 are more or less the same of lines 35-36-37
-line 81, please replace “feber” with “fever”
-line 152: choose between full stop or removing capital letter
-line 173: may be “January 7th, 2019”? Please verify.
Figure 1. This not work as clarification of your study, instead it is appeared messy. Take in consideration to revise it.
-line 214: it could be better “...was performed using....”
-line 272: remove full stop between pathogens and across
-Table 2: please adequate to the content the fourth and seventh column.
-line 329: close bracket somewhere.
-Tables 5 and 6: is it correct to use “Sig.” or better “P”?
-line 353: add a full stop at the end of the line
-line 464: please use Italy style for “Streptococcus zooepidemicus”
Author Response
All the team responsible for this paper acknowledge the comments from the reviewers and editor, as they help to improve the quality of our manuscript. In the following paragraphs, we will describe and address how referees’ new recommendations were followed. A point-by-point response to comments is provided as well as a file where changes are highlighted.
[1st Reviewer]
Comments and Suggestions for Authors
The authors presented a paper on serum amyloid A and pathogens qPCR levels in feral donkeys in California.
In my opinion the research is adequately conducted and the paper is well written.
I have only some minor comments to suggest before the acceptance for the publication.
Response: We thank the reviewer for his/her kind comments.
General comment: in my opinion the introduction section is too long and strong to read. The authors furnished a lot of unnecessary information about viruses and bacteria. Please try to summarize this section.
Response: Introduction section was summarised and reduced from 1433 to 1037 words.
More in details:
-if possible try to better differentiate the first lines in simple summary and abstract: lines 23-24-25 are more or less the same of lines 35-36-37
Response: We rewrote the first lines of simple summary to differentiate both sections.
-line 81, please replace “feber” with “fever”
Response: The paragraph was removed to shorten the introduction as suggested by the reviewer.
-line 152: choose between full stop or removing capital letter
Response: We added a full stop.
-line 173: may be “January 7th, 2019”? Please verify.
Response: We verified it and changed.
Figure 1. This not work as clarification of your study, instead it is appeared messy. Take in consideration to revise it.
Response: We deleted it from the manuscript.
-line 214: it could be better “...was performed using....”
Response: Changed.
-line 272: remove full stop between pathogens and across
Response: Removed.
-Table 2: please adequate to the content the fourth and seventh column.
Respons: Content was adequated.
-line 329: close bracket somewhere.
Response: Bracket was closed.
-Tables 5 and 6: is it correct to use “Sig.” or better “P”?
Response: Changed.
-line 353: add a full stop at the end of the line
Response: Added.
-line 464: please use Italy style for “Streptococcus zooepidemicus”
Response: Streptococcus zooepidemicus format was changed to italics.
Reviewer 2 Report
The authors performed observational assessments and collected blood and nasal swab samples from feral donkeys that had been recently caught and held at staging facilities. Body condition score (BCS) and other observations (behavior, lameness, skin lesions/wounds, nasal or ocular discharge, increased breathing effort and coughing) were scored and serum amyloid A (SAA) was measured on-site with a commercial device to detect evidence of systemic inflammation. Nasal swabs were tested by qPCR to detect cDNA for a panel of potential respiratory pathogens. The data collected are of interest and are worthy of publication in Animals. However, before the manuscript can be considered acceptable for publication, several issues will need to be addressed as well as substantial editing for content and grammatical errors. Overall, the manuscript is much too long and lack focus on the questions the authors are addressing. First, the Introduction is written somewhat as a review of donkey respiratory disease and is much too long. I encourage the authors to rewrite the Introduction, decreasing length by at least one half and focusing on their question of potential disease transmission by recently captured donkeys. Second, the authors provide data on physical characteristics (BCS) and observations yet the only scale they provide is for BCS (1-5). They direct the reader to “premises” in previous publications for the other scores. This is unacceptable and the authors should define the scoring systems – perhaps in a Table that could replace the unnecessary Figure 1. Third, the authors state that they have tested for viral and bacterial DNA by qPCR. Although qPCR does detect cDNA from RNA viruses (influenza and rhinovirus A and B), the way the authors have written the manuscript implies that all the viruses being tested for are DNA viruses – it should be made clear that the authors are screening for both RNA and DNA viruses. Next, Streptococcus equi subspecies zooepidemicus (“z” is not capitalized) is considered a opportunistic commensal of the airway of the horse, not a primary pathogen. Similarly, there is little evidence that AHV-2 is a pathogen of donkeys. The authors need to carefully state that they are not necessarily detecting pathogens – they are detecting viruses and bacteria. The authors may also want to include a recent study of a strangles outbreak on donkey breeding farms in China in this manuscript (https://doi.org/10.1111/evj.13114). Just because Streptococcus equi subspecies equi was not detected in this group of donkeys, it does not mean that all cohorts of captured donkeys will be similarly safe to comingle with other horses and donkeys. Fourth, the section on statistical methods is long and convoluted – and repeated throughout the Results. Please revise and shorten to make the section straightforward. Further, in the Results, there is little value in including Tables 3-6, in this reviewer’s opinion. Fifth, after attempting to distill through the data, it appears that these donkeys were overall fairly healthy at the time of initial capture. However, at the 30/60 day re-examination (apparently not all the same donkeys in the original Nov 4/5 and Dec 4/5 groups) respiratory sign of nasal discharge and cough were less (if I am reading Table 4 correctly) yet BCS had dropped by 0.69. A loss of BCS during the holding period would be a concerns and warrants further discussion. Sixth, there was minimal evidence of systemic inflammation, as assessed by SAA concentrations in whole blood (not full blood), in these cohorts of donkeys. Thus, it would seem somewhat senseless to measure SAA in future groups of captured donkeys, if further studies are pursued. The lack of an increase in SAA would further support that detection of AHV-2 and Streptococcus equi subspecies zooepidemicus was more indicative of a commensal organism than a pathogen. Further, if SAA had been substantially increased in some donkeys, especially younger animals, the authors would be unable to differentiate a potential increase due to a respiratory “pathogen” as compared to a burden of intestinal parasites. In fact, it may be of similar or even greater concern to consider the increased risk of intestinal parasitism, again especially in younger donkeys, after capture and relocation to facilities where stocking density would be higher. All in all, the authors have presented too much detail in this manuscript such that their question of interest is not clearly focused or elucidated. The manuscript requires an extensive rewrite. Further, if the primary author has English as a second language (suspected from the many grammatical errors), a revised manuscript should be critically reviewed (and edited) by other authors to ensure that it reads well and that grammar is correct throughout the manuscript. As an example, I have provided specific recommended changes for the ABSTRACT that are detailed as follows: ABSTRACT Line 36 – delete “and wellbeing” – not the focus of this sentence Lines 37-38 – simplify – change “the inflammatory markers (serum amyloid A) derived from the pathogenic processes that they involve” to “serum amyloid A” Line 39 – change “Blood and nasal swabs were sampled” to “Blood samples and nasal swabs were collected” Lines 42-43 – probably should list viruses before bacteria in the parentheses to be consistent with prior text; also influenza and rhinitis viruses are RNA viruses, not DNA viruses so qPCR should be restated as detecting genomic material or simply the viruses; [] should be used within () Lines 44-45 – should state that the relations were found ”in donkeys” from which AHV-2 and Strep were detected Lines 45-48 – you are saying the same thing twice in this first sentence – simplify; positive tests do not report – rephrase sentence with correct grammar Lines 48-50 – rephrase – one study is not “conclusive” rather the results support that donkeys do not appear to be a substantial risk for disease transmission to horses – but could be if they were carrying stranglesAuthor Response
All the team responsible for this paper acknowledge the comments from the reviewers and editor, as they help to improve the quality of our manuscript. In the following paragraphs, we will describe and address how referees’ new recommendations were followed. A point-by-point response to comments is provided as well as a file where changes are highlighted.
[2nd Reviewer]
Comments and Suggestions for Authors
The authors performed observational assessments and collected blood and nasal swab samples from feral donkeys that had been recently caught and held at staging facilities. Body condition score (BCS) and other observations (behavior, lameness, skin lesions/wounds, nasal or ocular discharge, increased breathing effort and coughing) were scored and serum amyloid A (SAA) was measured on-site with a commercial device to detect evidence of systemic inflammation. Nasal swabs were tested by qPCR to detect cDNA for a panel of potential respiratory pathogens. The data collected are of interest and are worthy of publication in Animals. However, before the manuscript can be considered acceptable for publication, several issues will need to be addressed as well as substantial editing for content and grammatical errors.
Response: Language in manuscript was reviewed by a Cambridge ESOL instructor in order to correct grammar and typos and to improve readability.
Overall, the manuscript is much too long and lack focus on the questions the authors are addressing.
Response: Manuscript was reduced from 8293 to 7626 words as requested by the reviewer.
First, the Introduction is written somewhat as a review of donkey respiratory disease and is much too long. I encourage the authors to rewrite the Introduction, decreasing length by at least one half and focusing on their question of potential disease transmission by recently captured donkeys.
Response: Introduction section was summarised to focus on potential disease transmission by recently captured donkeys and reduced from 1433 to 1037 words.
Second, the authors provide data on physical characteristics (BCS) and observations yet the only scale they provide is for BCS (1-5). They direct the reader to “premises” in previous publications for the other scores. This is unacceptable and the authors should define the scoring systems – perhaps in a Table that could replace the unnecessary Figure 1.
Response: Figure 1 was removed. A new figure depicting BCS was created to follow the suggestion by the reviewer. This BCS scale is globally accepted and has been widely described and published. We also present the references that we consulted to score BCS. We discarded providing a table, as the scale that we used has been frequently described in literature and given both reviewers suggested the manuscript was long, hence, to avoid extending the manuscript content unnecessarily.
Third, the authors state that they have tested for viral and bacterial DNA by qPCR. Although qPCR does detect cDNA from RNA viruses (influenza and rhinovirus A and B), the way the authors have written the manuscript implies that all the viruses being tested for are DNA viruses – it should be made clear that the authors are screening for both RNA and DNA viruses.
Response: We clarified the information as requested by reviewer.
Next, Streptococcus equi subspecies zooepidemicus (“z” is not capitalized) is considered a opportunistic commensal of the airway of the horse, not a primary pathogen.
Response: We checked and corrected the typo.
Similarly, there is little evidence that AHV-2 is a pathogen of donkeys. The authors need to carefully state that they are not necessarily detecting pathogens – they are detecting viruses and bacteria.
Response: We clarified across the text.
The authors may also want to include a recent study of a strangles outbreak on donkey breeding farms in China in this manuscript (https://doi.org/10.1111/evj.13114).
Response: We added a reference to this study.
Just because Streptococcus equi subspecies equi was not detected in this group of donkeys, it does not mean that all cohorts of captured donkeys will be similarly safe to comingle with other horses and donkeys.
Response: We clarified.
Fourth, the section on statistical methods is long and convoluted – and repeated throughout the Results. Please revise and shorten to make the section straightforward.
Response: We apologize for the way in which statistical section was written as it was very difficult to follow what we wanted to express. The section was rewritten completely.
Further, in the Results, there is little value in including Tables 3-6, in this reviewer’s opinion. Fifth, after attempting to distill through the data, it appears that these donkeys were overall fairly healthy at the time of initial capture. However, at the 30/60 day re-examination (apparently not all the same donkeys in the original Nov 4/5 and Dec 4/5 groups) respiratory sign of nasal discharge and cough were less (if I am reading Table 4 correctly) yet BCS had dropped by 0.69. A loss of BCS during the holding period would be a concerns and warrants further discussion.
Response: We disagree as these tables present the key results of the analyses tested, otherwise, which results would our conclusions be based on and we would remove the opportunity for readers to draw their own conclusions in the light of their results. For instance, as explained in the article Table 3 shows a picture of the data in our sample. This is necessary otherwise; the statistical context of the study is missing, that is readers do not know which was the scene that we found in the field. Table 4 shows the statistics that helped to determine whether a significant (relevant) difference in SAA concentrations and viral and bacterial DNA quantitates was found between sampling moment 1 and sampling moment 2. Table 5 tells us which percentage of the variability in the data of our study is linearly explained by the factors in Table 6, otherwise the models’ origin is blurred.
Sixth, there was minimal evidence of systemic inflammation, as assessed by SAA concentrations in whole blood (not full blood), in these cohorts of donkeys. Thus, it would seem somewhat senseless to measure SAA in future groups of captured donkeys, if further studies are pursued.
Response: As we expressed at the end of the manuscript our study evaluates SAA levels in healthy donkeys but additional research measuring SAA in clinically ill animals would be beneficial. In line with one of your previous comments in regards Streptococcus presence, the picture in this group may not be translatable to other groups, specially when SAA levels are connected to the presence of certain viruses or bacteria and their levels, which may vary, hence, further studies may still clarify trends in these parameters.
The lack of an increase in SAA would further support that detection of AHV-2 and Streptococcus equi subspecies zooepidemicus was more indicative of a commensal organism than a pathogen. Further, if SAA had been substantially increased in some donkeys, especially younger animals, the authors would be unable to differentiate a potential increase due to a respiratory “pathogen” as compared to a burden of intestinal parasites. In fact, it may be of similar or even greater concern to consider the increased risk of intestinal parasitism, again especially in younger donkeys, after capture and relocation to facilities where stocking density would be higher.
Response:
All in all, the authors have presented too much detail in this manuscript such that their question of interest is not clearly focused or elucidated. The manuscript requires an extensive rewrite. Further, if the primary author has English as a second language (suspected from the many grammatical errors), a revised manuscript should be critically reviewed (and edited) by other authors to ensure that it reads well and that grammar is correct throughout the manuscript.
Response: Grammar and typos were checked and corrected across the manuscript. A Cambridge ESOL examinations instructor checked the whole manuscript in order to correct potential grammar incongruencies and improve clarity.
As an example, I have provided specific recommended changes for the ABSTRACT that are detailed as follows: ABSTRACT Line 36 – delete “and wellbeing” – not the focus of this sentence
Response: Reviewer suggestion was followed.
Lines 37-38 – simplify – change “the inflammatory markers (serum amyloid A) derived from the pathogenic processes that they involve” to “serum amyloid A”
Response: Reviewer suggestion was followed.
Line 39 – change “Blood and nasal swabs were sampled” to “Blood samples and nasal swabs were collected”
Response: Reviewer suggestion was followed.
Lines 42-43 – probably should list viruses before bacteria in the parentheses to be consistent with prior text; also influenza and rhinitis viruses are RNA viruses, not DNA viruses so qPCR should be restated as detecting genomic material or simply the viruses; [] should be used within ()
Response: Reviewer suggestion was followed.
Lines 44-45 – should state that the relations were found ”in donkeys” from which AHV-2 and Strep were detected
Response: Reviewer suggestion was followed.
Lines 45-48 – you are saying the same thing twice in this first sentence – simplify; positive tests do not report – rephrase sentence with correct grammar
Response: Reviewer suggestion was followed.
Lines 48-50 – rephrase – one study is not “conclusive” rather the results support that donkeys do not appear to be a substantial risk for disease transmission to horses – but could be if they were carrying strangles
Response: Reviewer suggestion was followed.
Round 2
Reviewer 2 Report
The authors revised manuscript is somewhat improved. However, it is still much longer than necessary and could be further improved with another revision.
In the Methods, the authors list the visual examination categories they observed but only define the 1-5 scale for BCS. It is unclear how the other categories are categorized. For example, 1 is the worst for BCS but typically would be the least (best condition) for lameness. The scales must be defined for all categories. Similarly, the legends for Tables 2 and 3 need to be made clear as to these same concerns, as well as to what the values fo the PCR results represent. Tables should be able to be understood without needing to refer back to the text.
Next, the authors did not follow a suggested deletion of Tables 4-6 as these statistical values add little to the primary manuscript - they could be included as supplementary material.
As pointed out to the authors in my initial review, loss of body condition score between the two observation times seems to be an important finding in this study that could predispose congregated donkeys to increased risk of disease. This point would seem to warrant discussion.
I still recommend having the manuscript reviewed by someone with a somewhat harsh red pen - the important information could be presented in a length that would be only 50-70% of the current length of the manuscript.
Author Response
The authors revised manuscript is somewhat improved. However, it is still much longer than necessary and could be further improved with another revision.
Response: The manuscript has been shortened following the suggestion of the reviewer from 18 to 13 pages (70%), implementing a reduction of more than 382 lines.
In the Methods, the authors list the visual examination categories they observed but only define the 1-5 scale for BCS. It is unclear how the other categories are categorized. For example, 1 is the worst for BCS but typically would be the least (best condition) for lameness. The scales must be defined for all categories.
Response: Thank you for your suggestions we have included a scale per parameter assessed.
Line 598-610 further describe the scales used to visually evaluate such conditions as lameness, behavior, coughing, nasal and ocular discharge. Excluding behavior, the other parameters were scored as “1-present or 2- not present “by attending clinicians. Please keep in mind we were working with wild donkeys that had no prior experience being handled by people therefore we had to visually assess such conditions that often may warrant further examination such as a through lameness exam but that was simply not an option when working with wild animals. All observational scoring and assessments were conducted in a timely manner to decrease any additional stress of the donkeys being sampled for blood and nasal secretions.
Line 609-610 further explains how the body of the donkey from head to tail were visually evaluated for behavior by the veterinarians and specialist in donkey and mule behavior. Observations included observing the eyes, ears, nostril, head, neck, and overall body position and postures to evaluate behavior scored during this assessment. Again, we were working with feral/wild donkeys so observations had to be made in a timely manner and the least amount of stress was kept in mind when performing evaluations. All evaluators (veterinarians and equine scientists) have worked for years with donkeys in many countries and environments and are highly skilled and experienced to make assessments on all parameters included in this wellness check.
Similarly, the legends for Tables 2 and 3 need to be made clear as to these same concerns, as well as to what the values fo the PCR results represent. Tables should be able to be understood without needing to refer back to the text.
Response: Lines 782-87
Table 2 now reads, Summary of descriptive statistics for clinical examination signs, BCS (1-5), Behaviour signs (1-alert, 2-apathetic, 3- lethargic, 4-aggressive, 5- fearful, 6-withdrawn), Lesions (1-present, 2-not present), Lameness (1-present, 2- not present), Ocular discharge (1- present, 2- not present), Nasal discharge (1- present, 2- not present), Breathing (1-normal, 2-abnormal), coughing (1- present, 2-not present), SAA maximal concentration expressed in mg/L, qPCR-assay microbe loads (40 = zero viral or bacterial present, lower numbers indicate higher number of DNA found), and controls at sampling moment one in fifty three donkeys including jacks, jennies, and foals.
Lines 826-831
Table 3 now reads, Summary of descriptive statistics for clinical examination signs, BCS (1-5), Behaviour signs (1-alert, 2-apathetic, 3- lethargic, 4-aggressive, 5- fearful, 6-withdrawn), Lesions (1-present, 2-not present), Lameness (1-present, 2- not present), Ocular discharge (1- present, 2- not present), Nasal discharge (1- present, 2- not present), Breathing (1-normal, 2-abnormal), coughing (1- present, 2-not present), SAA maximal concentration expressed in mg/L, qPCR-assay microbe loads (40 = zero viral or bacterial presence, lower numbers indicate higher number of DNA found), and controls at sampling moment two in thirty donkeys (28 jennies and 4 foals) in long term holding facilities.
As pointed out to the authors in my initial review, loss of body condition score between the two observation times seems to be an important finding in this study that could predispose congregated donkeys to increased risk of disease. This point would seem to warrant discussion.
Response: Point noted and has been addressed. The change in the overall body condition score may have been related to the change in number of donkeys, primarily jacks in the first testing and evaluation who were not of course in foal or lactating. The second testing focused on donkeys at the long-term holding facility that were all jennies and some jennies were in foal and or nursing foals. The change in environment, temperature, relocation and change in diet likely lead to a change in BCS along with an increase in viral loads.
Line 1012-1013- Jennies who were relocated to long term holding facility and co-mingled with donkeys from multiple locations showed an increase in Asinine Herpesvirus 2, Streptococcus zooepidemicus and a decrease in the mean BCS. Granted, these jennies were lactating and nursing foals as well as changed environments and diets.
Next, the authors did not follow a suggested deletion of Tables 4-6 as these statistical values add little to the primary manuscript - they could be included as supplementary material.
Response: Tables 4 to 6 were included as supplementary material.
I still recommend having the manuscript reviewed by someone with a somewhat harsh red pen - the important information could be presented in a length that would be only 50-70% of the current length of the manuscript.
Response: Thank you for your suggestion to decrease the content in the current manuscript. We respect your opinion and we have now moved Tables 4-6 and the content associated with these tables to supplementary documents. The manuscript has been shortened following the suggestion of the reviewer from 16 to 13 pages (70%), implementing a reduction of more than 382 lines as suggested by the reviewer. To be considerate of the other reviewer who did not make such a recommendation, we have decreased the content without changing the message or intent of the manuscript and research findings. We hope this will suffice.
To this end, we thank you for your time and attention and for considering this manuscript again.